# Learning to solve TV regularised problems with unrolled algorithms

**Hamza Cherkaoui**
Université Paris-Saclay, CEA, Inria
Gif-sur-Yvette, 91190, France
`hamza.cherkaoui@cea.fr`

**Jeremias Sulam**
Johns Hopkins University
`jsulam1@jhu.edu`

**Thomas Moreau**
Université Paris-Saclay, Inria, CEA,
Palaiseau, 91120, France
`thomas.moreau@inria.fr`

## Abstract

Total Variation (TV) is a popular regularization strategy that promotes piece-wise constant signals by constraining the $\ell_1$-norm of the first order derivative of the estimated signal. The resulting optimization problem is usually solved using iterative algorithms such as proximal gradient descent, primal-dual algorithms or ADMM. However, such methods can require a very large number of iterations to converge to a suitable solution. In this paper, we accelerate such iterative algorithms by unfolding proximal gradient descent solvers in order to learn their parameters for 1D TV regularized problems. While this could be done using the *synthesis* formulation, we demonstrate that this leads to slower performances. The main difficulty in applying such methods in the *analysis* formulation lies in proposing a way to compute the derivatives through the proximal operator. As our main contribution, we develop and characterize two approaches to do so, describe their benefits and limitations, and discuss the regime where they can actually improve over iterative procedures. We validate those findings with experiments on synthetic and real data.

## 1 Introduction

Ill-posed inverse problems appear naturally in signal and image processing and machine learning, requiring extra regularization techniques. Total Variation (TV) is a popular regularization strategy with a long history (Rudin et al., 1992), and has found a large number of applications in neuro-imaging (Fikret et al., 2013), medical imaging reconstruction (Tian et al., 2011), among myriad applications (Rodríguez, 2013; Darbon and Sigelle, 2006). TV promotes piece-wise constant estimates by penalizing the $\ell_1$-norm of the first order derivative of the estimated signal, and it provides a simple, yet efficient regularization technique.

TV-regularized problems are typically convex, and so a wide variety of algorithms are in principle applicable. Since the $\ell_1$ norm in the TV term is non-smooth, Proximal Gradient Descent (PGD) is the most popular choice (Rockafellar, 1976). Yet, the computation for the corresponding proximal operator (denoted prox-TV) represents a major difficulty in this case as it does not have a closed-form analytic solution. For 1D problems, it is possible to rely on dynamic programming to compute prox-TV, such as the taut string algorithm (Davies and Kovac, 2001; Condat, 2013a). Another alternative consists in computing the proximal operator with iterative first order algorithm (Chambolle, 2004; Beck and Teboulle, 2009; Boyd et al., 2011; Condat, 2013b). Other algorithms to solve TV-regularized

problems rely on primal dual algorithms (Chambolle and Pock, 2011; Condat, 2013b) or Alternating Direction Method of Multipliers (ADMM) (Boyd et al., 2011). These algorithms typically use one sequence of estimates for each term in the objective and try to make them as close as possible while minimizing the associated term. While these algorithms are efficient for denoising problems – where one is mainly concerned with good reconstruction – they can result in estimate that are not very well regularized if the two sequences are not close enough.

When on fixed computational budget, iterative optimization methods can become impractical as they often require many iterations to give a satisfactory estimate. To accelerate the resolution of these problems with a finite (and small) number of iterations, one can resort to unrolled and learned optimization algorithms (see Monga et al. 2019 for a review). In their seminal work, Gregor and Le Cun (2010) proposed the Learned ISTA (LISTA), where the parameters of an unfolded Iterative Shrinkage-Thresholding Algorithm (ISTA) are learned with gradient descent and back-propagation. This allows to accelerate the approximate solution of a Lasso problem (Tibshirani, 1996), with a fixed number of iteration, for signals from a certain distribution. The core principle behind the success of this approach is that the network parameters can adaptively leverage the sensing matrix structure (Moreau and Bruna, 2017) as well as the input distribution (Giryes et al., 2018; Ablin et al., 2019). Many extensions of this original idea have been proposed to learn different algorithms (Sprechmann et al., 2012, 2013; Borgerding et al., 2017) or for different classes of problem (Xin et al., 2016; Giryes et al., 2018; Sulam et al., 2019). The motif in most of these adaptations is that all operations in the learned algorithms are either linear or separable, thus resulting in sub-differentials that are easy to compute and implement via back-propagation. Algorithm unrolling is also used in the context of bi-level optimization problems such as hyper-parameter selection. Here, the unrolled architecture provides a way to compute the derivative of the inner optimization problem solution compared to another variable such as the regularisation parameter using back-propagation (Bertrand et al., 2020).

The focus of this paper is to apply algorithm unrolling to TV-regularized problems in the 1D case. While one could indeed apply the LISTA approach directly to the *synthesis* formulation of these problems, we show in this paper that using such formulation leads to slower iterative or learned algorithms compared to their *analysis* counterparts. The extension of learnable algorithms to the analysis formulation is not trivial, as the inner proximal operator does not have an analytical or separable expression. We propose two architectures that can learn TV-solvers in their analysis form directly based on PGD. The first architecture uses an exact algorithm to compute the prox-TV and we derive the formulation of its weak Jacobian in order to learn the network's parameters. Our second method rely on a nested LISTA network in order to approximate the prox-TV itself in a differentiable way. This latter approach can be linked to inexact proximal gradient methods (Schmidt et al., 2011; Machart et al., 2012). These results are backed with numerical experiments on synthetic and real data. Concurrently to our work, Lecouat et al. (2020) also proposed an approach to differentiate the solution of TV-regularized problems. While their work can be applied in the context of 2D signals, they rely on smoothing the regularization term using Moreau-Yosida regularization, which results in smoother estimates from theirs learned networks. In contrast, our work allows to compute sharper signals but can only be applied to 1D signals.

The rest of the paper is organized as follows. In Section 2, we describe the different formulations for TV-regularized problems and their complexity. We also recall central ideas of algorithm unfolding. Section 3 introduces our two approaches for learnable network architectures based on PGD. Finally, the two proposed methods are evaluated on real and synthetic data in Section 4.

**Notations** For a vector $x \in \mathbb{R}^k$, we denote $\|x\|_q$ its $\ell_q$-norm. For a matrix $A \in \mathbb{R}^{m \times k}$, we denote $\|A\|_2$ its $\ell_2$-norm, which corresponds to its largest singular value and $A^\dagger$ denotes its pseudo-inverse. For an ordered subset of indices $\mathcal{S} \subset \{1, \ldots, k\}$, $x_\mathcal{S}$ denote the vector in $\mathbb{R}^{|\mathcal{S}|}$ with element $(x_\mathcal{S})_t = x_{i_t}$ for $i_t \in \mathcal{S}$. For a matrix $A \in \mathbb{R}^{m \times k}$, $A_{:,\mathcal{S}}$ denotes the sub-matrix $[A_{:,i_1}, \ldots A_{:,i_{|\mathcal{S}|}}]$ composed with the columns $A_{:,i_t}$ of index $i_t \in \mathcal{S}$ of $A$. For the rest of the paper, we refer to the operators $D \in \mathbb{R}^{k-1 \times k}, \widetilde{D} \in \mathbb{R}^{k \times k}, L \in \mathbb{R}^{k \times k}$ and $R \in \mathbb{R}^{k \times k}$ as:

$$D = \begin{bmatrix} -1 & 1 & 0 & \ldots & 0 \\ 0 & -1 & 1 & \ddots & \vdots \\ \vdots & \ddots & \ddots & \ddots & 0 \\ 0 & \ldots & 0 & -1 & 1 \end{bmatrix} \quad \widetilde{D} = \begin{bmatrix} 1 & 0 & \ldots & 0 \\ -1 & 1 & \ddots & \vdots \\ & \ddots & \ddots & 0 \\ & 0 & -1 & 1 \end{bmatrix} \quad L = \begin{bmatrix} 1 & 0 & \ldots & 0 \\ 1 & 1 & \ddots & \vdots \\ \vdots & \ddots & \ddots & 0 \\ 1 & \ldots & 1 & 1 \end{bmatrix} \quad R = \begin{bmatrix} 0 & 0 & \ldots & 0 \\ 0 & 1 & \ddots & \vdots \\ \vdots & \ddots & \ddots & 0 \\ 0 & \ldots & 0 & 1 \end{bmatrix}$$

## 2   Solving TV-regularized problems

We begin by detailing the TV-regularized problem that will be the main focus of our work. Consider a latent vector $u \in \mathbb{R}^k$, a design matrix $A \in \mathbb{R}^{m \times k}$ and the corresponding observation $x \in \mathbb{R}^m$. The original formulation of the TV-regularized regression problem is referred to as the *analysis* formulation (Rudin et al., 1992). For a given regularization parameter $\lambda > 0$, it reads

$$\min_{u \in \mathbb{R}^k} P(u) = \frac{1}{2} \|x - Au\|_2^2 + \lambda \|u\|_{TV}, \tag{1}$$

where $\|u\|_{TV} = \|Du\|_1$, and $D \in \mathbb{R}^{k-1 \times k}$ stands for the first order finite difference operator, as defined above. The problem in (1) can be seen as a special case of a Generalized Lasso problem (Tibshirani and Taylor, 2011); one in which the analysis operator is $D$. Note that problem $P$ is convex, but the $TV$-norm is non-smooth. In these cases, a practical alternative is the PGD, which iterates between a gradient descent step and the prox-TV. This algorithm's iterates read

$$u^{(t+1)} = \text{prox}_{\frac{\lambda}{\rho} \|\cdot\|_{TV}} \left( u^{(t)} - \frac{1}{\rho} A^\top (Au^{(t)} - x) \right) , \tag{2}$$

where $\rho = \|A\|_2^2$ and the prox-TV is defined as

$$\text{prox}_{\mu \|\cdot\|_{TV}} (y) = \arg\min_{u \in \mathbb{R}^k} F_y(u) = \frac{1}{2} \|y - u\|_2^2 + \mu \|u\|_{TV}. \tag{3}$$

Problem (3) does not have a closed-form solution, and one needs to resort to iterative techniques to compute it. In our case, as the problem is 1D, the prox-TV problem can be addressed with a dynamic programming approach, such as the taut-string algorithm (Condat, 2013a). This scales as $O(k)$ in all practical situations and is thus much more efficient than other optimization based iterative algorithms (Rockafellar, 1976; Chambolle, 2004; Condat, 2013b) for which each iteration is $O(k^2)$ at best.

With a generic matrix $A \in \mathbb{R}^{m \times k}$, the PGD algorithm is known to have a sublinear convergence rate (Combettes and Bauschke, 2011). More precisely, for any initialization $u^{(0)}$ and solution $u^*$, the iterates satisfy

$$P(u^{(t)}) - P(u^*) \leq \frac{\rho}{2t} \|u^{(0)} - u^*\|_2^2, \tag{4}$$

where $u^*$ is a solution of the problem in (1). Note that the constant $\rho$ can have a significant effect. Indeed, it is clear from (4) that doubling $\rho$ leads to consider doubling the number of iterations.

### 2.1   Synthesis formulation

An alternative formulation for TV-regularized problems relies on removing the analysis operator $D$ from the $\ell_1$-norm and translating it into a synthesis expression (Elad et al., 2007). Removing $D$ from the non-smooth term simplifies the expression of the proximal operator by making it separable, as in the Lasso. The operator $D$ is not directly invertible but keeping the first value of the vector $u$ allows for perfect reconstruction. This motivates the definition of the operator $\widetilde{D} \in \mathbb{R}^{k \times k}$, and its inverse $L \in \mathbb{R}^{k \times k}$, as defined previously. Naturally, $L$ is the discrete integration operator. Considering the change of variable $z = \widetilde{D}u$, and using the operator $R \in \mathbb{R}^{k \times k}$, the problem in (1) is equivalent to

$$\min_{z \in \mathbb{R}^k} S(z) = \frac{1}{2} \|x - ALz\|_2^2 + \lambda \|Rz\|_1. \tag{5}$$

Note that for any $z \in \mathbb{R}^k$, $S(z) = P(Lz)$. There is thus an exact equivalence between solutions from the synthesis and the analysis formulation, and the solution for the analysis can be obtained with $u^* = Lz^*$. The benefit of this formulation is that the problem above now reduces to a Lasso problem (Tibshirani, 1996). In this case, the PGD algorithm is reduced to the ISTA with a closed-form proximal operator (the soft-thresholding). Note that this simple formulation is only possible in 1D where the first order derivative space is unconstrained. In larger dimensions, the derivative must be constrained to verify the Fubini's formula that enforces the symmetry of integration over dimensions. While it is also possible to derive synthesis formulation in higher dimension (Elad et al., 2007), this does not lead to simplistic proximal operator.

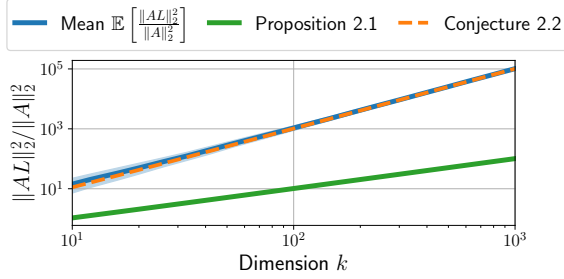

Figure 1: Evolution of $\mathbb{E}\left[\frac{\|AL\|_2^2}{\|A\|_2^2}\right]$ *w.r.t* the dimension $k$ for random matrices $A$ with *i.i.d* normally distributed entries. In light blue is the confidence interval [0.1, 0.9] computed with the quantiles. We observe that it scales as $O(k^2)$ and that our conjectured bound seems tight.

For this synthesis formulation, with a generic matrix $A \in \mathbb{R}^{m \times k}$, the PGD algorithm has also a sublinear convergence rate (Beck and Teboulle, 2009) such that

$$P(u^{(t)}) - P(u^*) \leq \frac{2\widetilde{\rho}}{t}\|u^{(0)} - u^*\|_2^2, \tag{6}$$

with $\widetilde{\rho} = \|AL\|_2^2$ (see Subsection F.1 for full derivation). While the rate of this algorithm is the same as in the analysis formulation – in $O(\frac{1}{t})$ – the constant $\widetilde{\rho}$ related to the operator norm differs. We now present two results that will characterize the value of $\widetilde{\rho}$.

**Proposition 2.1.** *[Lower bound for the ratio $\frac{\|AL\|_2^2}{\|A\|_2^2}$ expectation] Let $A$ be a random matrix in $\mathbb{R}^{m \times k}$ with i.i.d normally distributed entries. The expectation of $\|AL\|_2^2/\|A\|_2^2$ is asymptotically lower bounded when $k$ tends to $\infty$ by*

$$\mathbb{E}\left[\frac{\|AL\|_2^2}{\|A\|_2^2}\right] \geq \frac{2k+1}{4\pi^2} + o(1)$$

The full proof can be found in Subsection F.3. The lower bound is constructed by using $A^T A \succeq \|A\|_2^2 u_1 u_1^\top$ for a unit vector $u_1$ and computing explicitely the expectation for rank one matrices. To assess the tightness of this bound, we evaluated numerically $\mathbb{E}\left[\frac{\|AL\|_2^2}{\|A\|_2^2}\right]$ on a set of 1000 matrices sampled with *i.i.d* normally distributed entries. The results are displayed *w.r.t* the dimension $k$ in Figure 1. It is clear that the lower bound from Proposition 2.1 is not tight. This is expected as we consider only the leading eigenvector of $A$ to derive it in the proof. The following conjecture gives a tighter bound.

**Conjecture 2.2** (Expectation for the ratio $\frac{\|AL\|_2^2}{\|A\|_2^2}$). *Under the same conditions as in Proposition 2.1, the expectation of $\|AL\|_2^2/\|A\|_2^2$ is given by*

$$\mathbb{E}\left[\frac{\|AL\|_2^2}{\|A\|_2^2}\right] = \frac{(2k+1)^2}{16\pi^2} + o(1) \ .$$

We believe this conjecture can potentially be proven with analogous developments as those in Proposition 2.1, but integrating over all dimensions. However, a main difficulty lies in the fact that integration over all eigenvectors have to be carried out jointly as they are not independent. This is subject of current ongoing work.

Finally, we can expect that $\widetilde{\rho}/\rho$ scales as $\Theta(k^2)$. This leads to the observation that $\frac{\widetilde{\rho}}{2} \gg \rho$ in large enough dimension. As a result, the analysis formulation should be much more efficient in terms of iterations than the synthesis formulation – as long as the prox-TVcan be dealt with efficiently.

## 2.2 Unrolled iterative algorithms

As shown by Gregor and Le Cun (2010), ISTA is equivalent to a recurrent neural network (RNN) with a particular structure. This observation can be generalized to PGD algorithms for any penalized least squares problem of the form

$$u^*(x) = \arg\min_u \mathcal{L}(x, u) = \frac{1}{2}\|x - Bu\|_2^2 + \lambda g(u) \ , \tag{7}$$

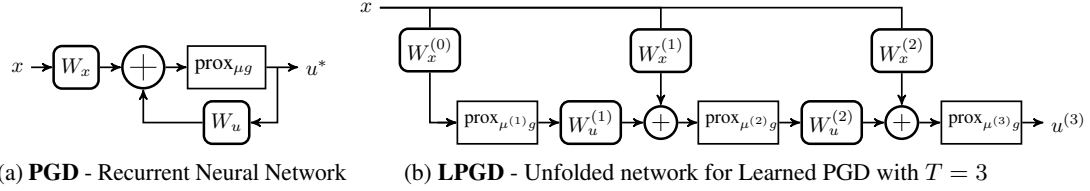

(a) **PGD** - Recurrent Neural Network   (b) **LPGD** - Unfolded network for Learned PGD with $T = 3$

Figure 2: **Algorithm Unrolling** - Neural network representation of iterative algorithms. The parameters $\Theta^{(t)} = \{W_x^{(t)}, W_u^{(t)}, \mu^{(t)}\}$ can be learned by minimizing the loss (10) to approximate good solution of (7) on average.

where $g$ is proper and convex, as depicted in Figure 2a. By unrolling this architecture with $T$ layers, we obtain a network $\phi_{\Theta^{(T)}}(x) = u^{(T)}$ – illustrated in Figure 2b – with parameters $\Theta^{(T)} = \{W_x^{(t)}, W_u^{(t)}, \mu^{(t)}\}_{t=1}^{T}$, defined by the following recursion

$$u^{(0)} = B^{\dagger}x \; ; \qquad u^{(t)} = \text{prox}_{\mu^{(t)}g}(W_x^{(t)}x + W_u^{(t)}u^{(t-1)}) \; . \tag{8}$$

As underlined by (4), a good estimate $u^{(0)}$ is crucial in order to have a fast convergence toward $u^*(x)$. However, this chosen initialization is mitigated by the first layer of the network which learns to set a good initial guess for $u^{(1)}$. For a network with $T$ layers, one recovers exactly the $T$-th iteration of PGD if the weights are chosen constant equal to

$$W_x^{(t)} = \frac{1}{\rho}B^{\top}, \qquad W_u^{(t)} = (\text{Id} - \frac{1}{\rho}B^{\top}B) \; , \qquad \mu^{(t)} = \frac{\lambda}{\rho}, \qquad \text{with } \rho = \|B\|_2^2 \; . \tag{9}$$

In practice, this choice of parameters are used as initialization for a posterior training stage. In many practical applications, one is interested in minimizing the loss (7) for a fixed $B$ and a particular distribution over the space of $x$, $\mathcal{P}$. As a result, the goal of this training stage is to find parameters $\Theta^{(T)}$ that minimize the risk, or expected loss, $\mathbb{E}[\mathcal{L}(x, \phi_{\Theta^{(T)}}(x))]$ over $\mathcal{P}$. Since one does not have access to this distribution, and following an empirical risk minimization approach with a given training set $\{x_1, \ldots x_N\}$ (assumed sampled *i.i.d* from $\mathcal{P}$), the network is trained by minimizing

$$\min_{\Theta^{(T)}} \frac{1}{N} \sum_{i=1}^{N} \mathcal{L}(x_i, \phi_{\Theta^{(T)}}(x_i)) \; . \tag{10}$$

Note that when $T \to +\infty$, the presented initialization in (9) gives a global minimizer of the loss for all $x_i$, as the network converges to exact PGD. When $T$ is fixed, however, the output of the network is not a minimizer of (7) in general. Minimizing this empirical risk can therefore find a weight configuration that reduces the sub-optimality of the network relative to (7) over the input distribution used to train the network. In such a way, the network learns an algorithm to approximate the solution of (7) for a particular class or distributions of signals. It is important to note here that while this procedure can accelerate the resolution the problem, the learned algorithm will only be valid for inputs $x_i$ coming from the same input distribution $\mathcal{P}$ as the training samples. The algorithm might not converge for samples which are too different from the training set, unlike the iterative algorithm which is guaranteed to converge for any sample.

This network architecture design can be directly applied to TV regularised problems if the synthesis formulation (5) is used. Indeed, in this case PGD reduces to the ISTA algorithm, with $B = AL$ and $\text{prox}_{\mu g} = \text{ST}(\cdot, \mu)$ becomes simply a soft-thresholding operator (which is only applied on the coordinates $\{2, \ldots k\}$, following the definition of $R$). However, as discussed in Proposition 2.1, the conditioning of the synthesis problem makes the estimation of the solution slow, increasing the number of network layers needed to get a good estimate of the solution. In the next section, we will extend these learning-based ideas directly to the analysis formulation by deriving a way to obtain exact and approximate expressions for the sub-differential of the non-separable prox-TV.

## 3 Back-propagating through TV proximal operator

Our two approaches to define learnable networks based on PGD for TV-regularised problems in the analysis formulation differ on the computation of the prox-TV and its derivatives. Our first approach

consists in directly computing the weak derivatives of the exact proximal operator while the second one uses a differentiable approximation.

## 3.1 Derivative of prox-TV

While there is no analytic solution to the prox-TV, it can be computed exactly (numerically) for 1D problems using the taut-string algorithm (Condat, 2013a). This operator can thus be applied at each layer of the network, reproducing the architecture described in Figure 2b. We define the LPGD-Taut network $\phi_{\Theta^{(T)}}(x)$ with the following recursion formula

$$\phi_{\Theta^{(T)}}(x) = \text{prox}_{\mu^{(T)}\|\cdot\|_{TV}}\left(W_x^{(T)}x + W_u^{(T)}\phi_{\Theta^{(T-1)}}(x)\right) \tag{11}$$

To be able to learn the parameters through gradient descent, one needs to compute the derivatives of (10) *w.r.t* the parameters $\Theta^{(T)}$. Denoting $h = W_x^{(t)}x + W_u^{(t)}\phi_{\Theta^{(t-1)}}(x)$ and $u = \text{prox}_{\mu^{(t)}\|\cdot\|_{TV}}(h)$, the application of the chain rule (as implemented efficiently by automatic differentiation) results in

$$\frac{\partial\mathcal{L}}{\partial h} = J_x(h,\mu^{(t)})^\top\frac{\partial\mathcal{L}}{\partial u}, \quad\text{and}\quad \frac{\partial\mathcal{L}}{\partial\mu^{(t)}} = J_\mu(h,\mu^{(t)})^\top\frac{\partial\mathcal{L}}{\partial u}, \tag{12}$$

where $J_x(h,\mu) \in \mathbb{R}^{k\times k}$ and $J_\mu(h,\mu) \in \mathbb{R}^{k\times 1}$ denotes the weak Jacobian of the output of the proximal operator $u$ with respect to the first and second input respectively. We now give the analytic formulation of these weak Jacobians in the following proposition.

**Proposition 3.1.** *[Weak Jacobian of prox-TV] Let $x \in \mathbb{R}^k$ and $u = prox_{\mu\|\cdot\|_{TV}}(x)$, and denote by $\mathcal{S}$ the support of $z = \widetilde{D}u$. Then, the weak Jacobian $J_x$ and $J_\mu$ of the prox-TV relative to $x$ and $\mu$ can be computed as*

$$J_x(x,\mu) = L_{:,\mathcal{S}}(L_{:,\mathcal{S}}^\top L_{:,\mathcal{S}})^{-1}L_{:,\mathcal{S}}^\top \quad\text{and}\quad J_\mu(x,\mu) = -L_{:,\mathcal{S}}(L_{:,\mathcal{S}}^\top L_{:,\mathcal{S}})^{-1}\text{sign}(Du)_{\mathcal{S}}$$

The proof of this proposition can be found in Subsection G.1. Note that the dependency in the inputs is only through $\mathcal{S}$ and $\text{sign}(Du)$, where $u$ is a short-hand for $\text{prox}_{\mu\|\cdot\|_{TV}}(x)$. As a result, computing these weak Jacobians can be done efficiently by simply storing $\text{sign}(Du)$ as a mask, as it would be done for a RELU or the soft-thresholding activations, and requiring just $2(k-1)$ bits. With these expressions, it is thus possible to compute gradient relatively to all parameters in the network, and employ them via back-propagation.

## 3.2 Unrolled prox-TV

As an alternative to the previous approach, we propose to use the LISTA network to approximate the prox-TV (3). The prox-TV can be reformulated with a synthesis approach resulting in a Lasso *i.e.*

$$z^* = \arg\min_z \frac{1}{2}\|h - Lz\|_2^2 + \mu\|Rz\|_1 \tag{13}$$

The proximal operator solution can then be retrieved with $\text{prox}_{\mu\|\cdot\|_{TV}}(h) = Lz^*$. This problem can be solved using ISTA, and approximated efficiently with a LISTA network Gregor and Le Cun (2010). For the resulting architecture – dubbed LPGD-LISTA – $\text{prox}_{\mu\|\cdot\|_{TV}}(h)$ is replaced by a nested LISTA network with a fixed number of layers $T_{in}$ defined recursively with $z^{(0)} = Dh$ and

$$z^{(\ell+1)} = \text{ST}\left(W_z^{(\ell,t)}z^{(\ell)} + W_h^{(\ell,t)}\Phi_{\Theta^{(t)}}, \frac{\mu^{(\ell,t)}}{\rho}\right). \tag{14}$$

Here, $W_z^{(\ell,t)}, W_h^{(\ell,t)}, \mu^{(\ell,t)}$ are the weights of the nested LISTA network for layer $\ell$. They are initialized with weights chosen as in (9) to ensure that the initial state approximates the prox-TV. Note that the weigths of each of these inner layers are also learned through back-propagation during training.

The choice of this architecture provides a differentiable (approximate) proximal operator. Indeed, the LISTA network is composed only of linear and soft-thresholding layers – standard tools for deep-learning libraries. The gradient of the network's parameters can thus be computed using classic automatic differentiation. Moreover, if the inner network is not trained, the gradient computed with this method will converge toward the gradient computed using Proposition 3.1 as $T_{in}$ goes to $\infty$ (see Proposition G.2). Thus, in this untrained setting with infinitely many inner layers, the network is equivalent to LPGD-Taut as the output of the layer also converges toward the exact proximal operator.

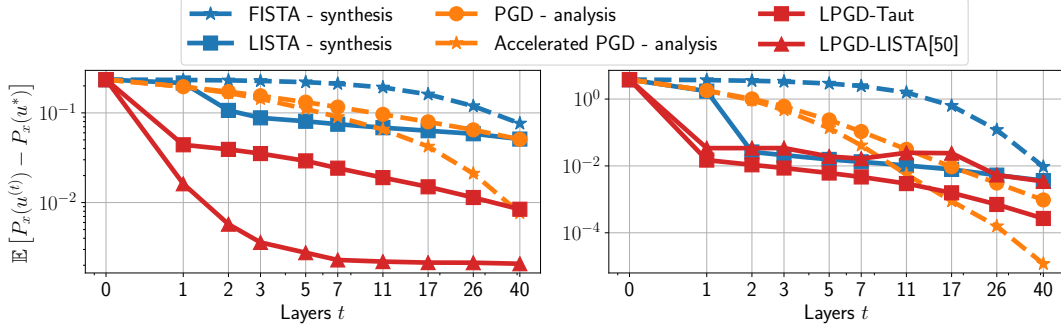

Figure 3: **Performance comparison** for different regularisation levels (*left*) $\lambda = 0.1$, (*right*) $\lambda = 0.8$. We see that synthesis formulations are outperformed by the analysis counter part. Both our methods are able to accelerate the resolution of (20), at least in the first iterations.

**Connections to inexact PGD**   A drawback of approximating the prox-TV via an iterative procedure is, precisely, that it is not exact. This optimization error results from a trade-off between computational cost and convergence rate. Using results from Machart et al. (2012), one can compute the scaling of $T$ and $T_{in}$ to reach an error level of $\delta$ with an untrained network. Proposition G.3 shows that without learning, $T$ should scale as $O(\frac{1}{t})$ and $T_{in}$ should be larger than $O(\ln(\frac{1}{\delta}))$. This scaling gives potential guidelines to set these parameters, as one can expect that learning the parameters of the network would reduce these requirement.

## 4   Experiments

All experiments are performed in Python using `PyTorch` (Paszke et al., 2019). We used the implementation[1] of Barbero and Sra (2018) to compute TV proximal operator using taut-string algorithm. The code to reproduce the figures is available online[2].

In all experiments, we initialize $u_0 = A^\dagger x$. Moreover, we employed a normalized $\lambda_{reg}$ as a penalty parameter: we first compute the value of $\lambda_{\max}$ (which is the minimal value for which $z = 0$ is solution of (5)) and we refer to $\lambda$ as the ratio so that $\lambda_{reg} = \lambda\lambda_{\max}$, with $\lambda \in [0, 1]$ (see Appendix D). As the computational complexity of all compared algorithms is the same except for the proximal operator, we compare them in term of iterations.

### 4.1   Simulation

We generate $n = 2000$ times series and used half for training and other half for testing and comparing the different algorithms. We train all the network's parameters jointly – those to approximate the gradient for each iteration along with those to define the inner proximal operator. The full training process is described in Appendix A. We set the length of the source signals $(u_i)_{i=1}^n \in \mathbb{R}^{n \times k}$ to $k = 8$ with a support of $|S| = 2$ non-zero coefficients (larger dimensions will be showcased in the real data application). We generate $A \in \mathbb{R}^{m \times k}$ as a Gaussian matrix with $m = 5$, obtaining then $(u_i)_{i=1}^n \in \mathbb{R}^{n \times p}$. Moreover, we add Gaussian noise to measurements $x_i = Au_i$ with a signal to noise ratio (SNR) of 1.0.

We compare our proposed methods, LPGD-Taut network and the LPGD-LISTA with $T_{in} = 50$ inner layers to PGD and Accelerated PGD with the analysis formulation. For completeness, we also add the FISTA algorithm for the synthesis formulation in order to illustrate Proposition 2.1 along with its learned version.

Figure 3 presents the risk (or expected function value, $P$) of each algorithm as a function of the number of layers or, equivalently, iterations. For the learned algorithms, the curves in $t$ display the performances of a network with $t$ layer trained specifically. We observe that all the synthesis formulation algorithms are slower than their analysis counterparts, empirically validating Proposition 2.1.

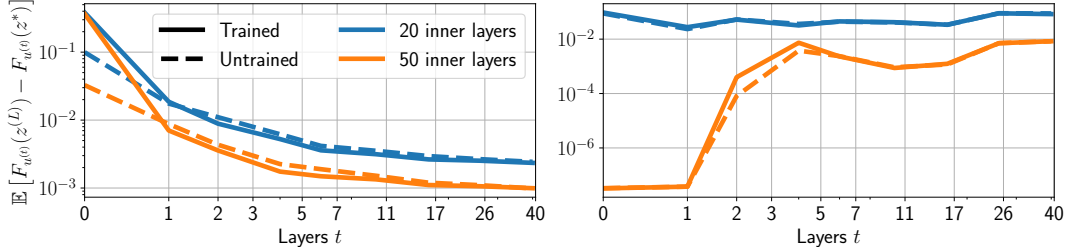

Figure 4: **Proximal operator error comparison** for different regularisation levels (*left*) $\lambda = 0.1$, (*right*) $\lambda = 0.8$. We see that learn the trained unrolled prox-TV barely improve the performance. More interestingly, in a high sparsity context, after a certain point, the error sharply increase.

Moreover, both of the proposed methods accelerate the resolution of (20) in a low iteration regime. However, when the regularization parameter is high ($\lambda = 0.8$), we observe that the performance of the LPGD-LISTA tends to plateau. It is possible that such a high level of sparsity require more than 50 layers for the inner network (which computes the prox-TV). According to Section 3.2, the error associated with this proximity step hinders the global convergence, making the loss function decrease slowly. Increasing the number of inner layers would alleviate this issue, though at the expense of increased computational burden for both training and runtime. For LPGD-Taut, while the Taut-string algorithm ensures that the recovered support is exact for the proximal step, the overall support can be badly estimated in the first iterations. This can lead to un-informative gradients as they greatly depend on the support of the solution in this case, and explain the reduced performances of the network in the high sparsity setting.

**Inexact prox-TV**   With the same data $(x_i)_{i=1}^n \in \mathbb{R}^{n \times m}$, we empirically investigate the error of the prox-TV $\epsilon_k^{(t)} = F_{u^{(t)}}(z^{(t)}) - F_{u^{(t)}}(z^*)$ and evaluate it for c with different number of layers ($T \in [20, 50]$). We also investigate the case where the parameter of the nested LISTA in LPGD-LISTA are trained compared to their initialization in untrained version.

Figure 4 depicts the error $\epsilon_k$ for each layer. We see that learning the parameters of the unrolled prox-TV in LPGD-LISTA barely improves the performance. More interestingly, we observe that in a high sparsity setting the error sharply increases after a certain number of layers. This is likely cause by the high sparsity of the estimates, the small numbers of iterations of the inner network (between 20 and 50) are insufficient to obtain an accurate solution to the proximal operator. This is in accordance with inexact PGD theory which predict that such algorithm has no exact convergence guarantees (Schmidt et al., 2011).

### 4.2  fMRI data deconvolution

Functional magnetic resonance imaging (fMRI) is a non-invasive method for recording the brain activity by dynamically measuring blood oxygenation level-dependent (BOLD) contrast, denoted here $x$. The latter reflects the local changes in the deoxyhemoglobin concentration in the brain Ogawa et al. (1992) and thus indirectly measures neural activity through the neurovascular coupling. This coupling is usually modelled as a linear and time-invariant system and characterized by its impulse response, the so-called haemodynamic response function (HRF), denoted here $h$. Recent developments propose to estimate either the neural activity signal independently (Fikret et al., 2013; Cherkaoui et al., 2019b) or jointly with the HRF (Cherkaoui et al., 2019a; Farouj et al., 2019). Estimating the neural activity signal with a fixed HRF is akin to a deconvolution problem regularized with TV-norm,

$$\min_{u \in \mathbb{R}^k} P(u) = \frac{1}{2}\|h * u - x\|_2^2 + \lambda \|u\|_{TV} \tag{15}$$

To demonstrate the usefulness of our approach with real data, where the training set has not the exact same distribution than the testing set, we compare the LPGD-Taut to Accelerated PGD for the analysis formulation on this deconvolution problem. We choose two subjects from the UK Bio Bank (UKBB) dataset (Sudlow et al., 2015), perform the usual fMRI processing and reduce the dimension of the problem to retain only 8000 time-series of 250 time-frames, corresponding to a record of 3 minute 03 seconds. The full preprocessing pipeline is described in Appendix B. We train

the LPGD taut-string network solver on the first subject and Figure 5 reports the performance of the two algorithms on the second subject for $\lambda = 0.1$. The performance is reported relatively to the number of iteration as the computational complexity of each iteration or layer for both methods is equivalent. It is clear that LPGD-Taut converges faster than the Accelerated PGD even on real data. In particular, acceleration is higher when the regularization parameter $\lambda$ is smaller. As mentioned previously, this acceleration is likely to be caused by the better learning capacity of the network in a low sparsity context. The same experiment is repeated for $\lambda = 0.8$ in Figure C.1.

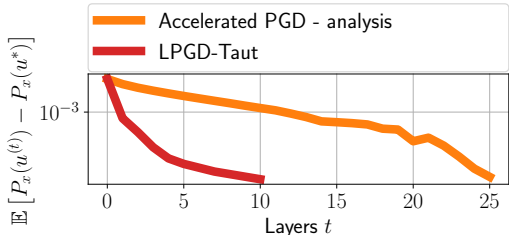

Figure 5: **Performance comparison** ($\lambda = 0.1$) between our analytic prox-TV derivative method and the PGD in the analysis formulation for the HRF deconvolution problem with fMRI data. Our proposed method outperform the FISTA algorithm in the analysis formulation.

## 5 Conclusion

This paper studies the optimization of TV-regularised problems via learned PGD. We demonstrated, both analytically and numerically, that it is better to address these problems in their original analysis formulation rather than resort to the simpler (alas slower) synthesis version. We then proposed two different algorithms that allow for the efficient computation and derivation of the required prox-TV, exactly or approximately. Our experiments on synthetic and real data demonstrate that our learned networks for prox-TV provide a significant advantage in convergence speed.

Finally, we believe that the principles presented in this paper could be generalized and deployed in other optimization problems, involving not just the TV-norm but more general analysis-type priors. In particular, this paper only apply for 1D TV problems because the equivalence between Lasso and TV is not exact in higher dimension. In this case, we believe exploiting a dual formulation (Chambolle, 2004) for the problem could allow us to derive similar learnable algorithms.

## Broader Impact

This work attempts to shed some understanding into empirical phenomena in signal processing – in our case, piecewise constant approximations. As such, it is our hope that this work encourages fellow researchers to invest in the study and development of principled machine learning tools. Besides these, we do not foresee any other immediate societal consequences.

## Acknowledgement

We gratefully acknowledge discussions with Pierre Ablin, whose suggestions helped us completing some parts of the proofs. H. Cherkaoui is supported by a CEA PhD scholarship. J. Sulam is partially supported by NSF Grant 2007649.

## Footnotes

[1]Available at `https://github.com/albarji/proxTV`

[2]Available at `https://github.com/hcherkaoui/carpet`.

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
