[Supplementary Material]

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

# A    Network training process strategy

Here, we give a more detailed description of the training procedure used in Section 4.

**Optimization algorithm for training**    In our experiments, all networks are trained using Gradient Descent (GD) with back-tracking line search. The gradients are computed using automatic differentiation in Pytorch (Paszke et al., 2019) for most layers and the weak Jacobian proposed in Subsection 3.1 for the back-propagation through the prox-TV. The learning is stopped once a step-size of $\eta_{limit} = 10^{-20}$ is reached in the back-tracking step. For LPGD-LISTA, the weights of the inner LISTA computing the prox-TV are trained jointly with the parameters of the outer unrolled PGD.

**Weight initialization**    All layers for an unrolled algorithm are initialized using the values of weights in (9) that ensure the output of the layer with $T$ layers corresponds to the output of $T$ iterations of the original algorithm. To further stabilize the training, we use a layer-wise approach. When training a network with $T_1 + T_2$ layers after having trained a network with $T_1$ layers, the first $T_1$ layers in the new network are initialized with the weights of the one trained previously, and the remaining layers are initialized using weights value from (9). This ensures that the initial value of the loss for the new network is smaller than the one from the shallower one if the unrolled algorithm is monotonous (as it is the case for PGD).

# B    Real fMRI data acquisition parameters and preprocessing strategy

In this section, we complete the description of the resting-state fMRI (rs-fMRI) data used for the experiment of Fig. 5. For this experiment, we investigate the 6 min long rs-fMRI acquisition (TR=0.735 s) from the UK Bio Bank dataset (Sudlow et al., 2015). The following pre-processing steps were applied on the images: motion correction, grand-mean intensity normalisation, high-pass temporal filtering, Echo planar imaging unwarping, Gradient Distortion Correction unwarping and structured artefacts removal by Independant Components Analysis. More details on the processing pipeline can found in Alfaro-Almagro et al. (2018).

On top of this preprocessing, we perform a standard fMRI preprocessing proposed in the python package Nilearn[3]. This standard pipeline includes to detrend the data, standardize it and filter high and low frequencies to reduce the presence of noise.

# C    Real fMRI data experiment addition results

Here, we provide an extra experiment for Subsection 4.2 with $\lambda = 0.8\lambda_{\max}$ and recall the previous one with $\lambda = 0.1\lambda_{\max}$ to help performance comparison in different regularization regime.

(a) $\lambda = 0.1\lambda_{\max}$                   (b) $\lambda = 0.8\lambda_{\max}$

Figure C.1: **Performance comparison** between LPGD-Taut and iterative PGD for the analysis formulation for the HRF deconvolution problem with fMRI data. Our proposed method outperform the FISTA algorithm in the analysis formulation. We notice a slight degradation of the acceleration in this high sparsity context.

We can see that the performance drop in the higher sparsity setting compared to the performances with $\lambda = 0.1\lambda_{\max}$ but LPGD-Taut still outperforms iterative algorithms for this task on real data.

## D    Computing $\lambda_{\max}$ for the TV regularized problem

The definition of $\lambda_{\max}$ is the smallest value for the regularisation parameter $\lambda$ such that the solution of the $TV$-regularized problem is constant. This corresponds to the definition of $\lambda_{\max}$ in the Lasso, which is the smallest regularisation parameter such that $0$ is solution. We here derive its analytic value which is used to rescale all experiments. This is important to define an equiregularisation set for the training and testing samples, to have a coherent and generalizable training.

**Proposition D.1.** *The value of $\lambda_{\max}$ for the TV-regularized problem is*

$$\lambda_{\max} = \|A^\top(Ac\mathbf{1} - x)\|_\infty$$

*where $c = \dfrac{\sum_{i=1}^{p} S_i x_i}{\sum_{i=1}^{p} S_i^2}$ and $S_i = \sum_{j=1}^{k} A_{i,j}$.*

*Proof.* We first derive the constant solution of the $\ell_2$-regression problem associated to (1). For $c \in \mathbb{R}$, we consider a constant vector $c\mathbf{1}$. The best constant solution for the $\ell_2$-regression problem is obtained by solving

$$\min_{c \in \mathbb{R}} f_x(c) = \frac{1}{2}\|x - cA\mathbf{1}\|_2^2 \ . \tag{16}$$

The first order optimality condition in c reads

$$\nabla f_x(c) = \sum_{i=1}^{n}(\sum_{j=1}^{k} A_{i,j})(c \sum_{j=1}^{k} A_{i,j} - x_i) = \sum_{i=1}^{n} S_i(cS_i - x_i) = 0 \ , \tag{17}$$

and thus $c = \dfrac{\sum_{i=1}^{p} S_i x_i}{\sum_{i=1}^{p} S_i^2}$.

Then, we look at the conditions on $\lambda$ to ensure that the constant solution $c\mathbf{1}$ is solution of the regularized problem. The first order conditions on the regularized problem reads

$$0 \in \partial P_x(c\mathbf{1}) = A^\top(Ac\mathbf{1} - x) + \lambda\partial\|Dc\mathbf{1}\|_1 \tag{18}$$

Next, we develop the previous equality:

$$\forall j \in \{2, \dots k\}, \quad A_j^\top(Ac\mathbf{1} - x) \in \lambda\partial(\|Dc\mathbf{1}\|_1)_j = [-\lambda, \lambda] \quad \text{since } Dc\mathbf{1} = 0 \tag{19}$$

Thus, the constrains are all satisfied for $\lambda \geq \lambda_{\max}$, with $\lambda_{max} = \|A^\top(Ac\mathbf{1} - x)\|_\infty$ and as $c$ is solution for the unregularized problem reduced to a constant, $c\mathbf{1}$ is solution of the TV-regularized problem for all $\lambda \geq \lambda_{\max}$.

$\square$

## E    Dual formulation

In this work, we devote our effort in the analysis formulation depicted in (1). In this section, we propose to investigate the dual formulation corresponding to (1) in order to rationalize our choice to focus on approaches that solve the prox-TV with an iterative method.

**Dual derivation**    First, we derive the dual of the analysis formulation for the prox-TV.

**Proposition E.1.** *[Dual re-parametrization for the analysis formulation TV problem (1)]*

*Considering the primal analysis problem (with operator and variables defined as previously)*

$$P_x(u) = \frac{1}{2}\|x - Au\|_2^2 + \lambda\|Du\|_1 \tag{20}$$

*Then, the dual formulation reads:*

$$p = -\min_{v} \frac{1}{2}\|{A^{\dagger}}^{\top} D^{\top} v\|_2^2 - v^{\top} D A^{\dagger} x \tag{21}$$

$$\text{s.t. } \|v\|_{\infty} \leq \lambda \tag{22}$$

*Proof.* Defining, $f$ and $g$, such as $f(u) = \frac{1}{2}\|x - Au\|_2^2$ and $g(u) = \lambda\|u\|_1$ and by denoting $p$ the minimum of (20) *w.r.t* $u$, the problem reads:

$$p = \min_{u} \quad f(u) + g(Du) \tag{23}$$

With the Fenchel-Rockafellar duality theorem, we derive the dual re-parametrization:

$$p = -\min_{v} \ f^*(-D^{\top}v) + g^*(v) \tag{24}$$

Note, that in this case we have the equality with $p$ since the problem (1) is $\mu$-strongly convex, with $\mu = \frac{1}{2}$.

We have $g^*(v) = -\min_u \ g(u) - v^{\top} u$. With a component-wise minimization, we obtain $g^*(v)_i = \delta_{|v_i| \leq \lambda}$ with $\delta$ being the convex indicator. Thus, we deduce that $g^*(v) = \delta_{\|v\|_{\infty} \leq \lambda}$.

Then, we have $f^*(v) = -\min_u \ f(u) - v^{\top} u$. By cancelling the gradient we obtain: $f^*(v) = \frac{1}{2}\|{A^{\dagger}}^{\top} v\|_2^2 + v^{\top} A^{\dagger} x$

This allows use to conclude the demonstration. Note that, if we set $A = \text{Id}$, we obtain the same problem as (Barbero and Sra, 2018; Chambolle, 2004).

$\square$

**Performance comparison**    We propose to compare the performance of different iterative solvers to assess their performance.

We generate $n = 1000$ times series to compare the performance between the different algorithms. We set the length of the source signals $(u_i)_{i=1}^n \in \mathbb{R}^{n \times k}$ to $k = 40$ with a support of $|S| = 4$ non-zero coefficients. We generate $A \in \mathbb{R}^{m \times k}$ as a Gaussian matrix with $m = 40$, obtaining then $(u_i)_{i=1}^n \in \mathbb{R}^{n \times p}$. Moreover, we add Gaussian noise to measurements $x_i = Au_i$ with a signal to noise ratio (SNR) of 1.0.

We select the PGD and its accelerated version with the synthesis primal formulation (5) ("Synthesis primal A/PGD"), the PGD and its accelerated version with the analysis primal formulation ("Analysis primal A/PGD"). We consider also the PGD and its accelerated version (Chambolle, 2004), for the analysis dual formulation ("Analysis dual A/PGD") and finally we add the primal/dual algorithm (Condat, 2013b) for the analysis primal formulation ("Analysis primal dual GD").

Figure E.1 a proposes performance comparison for an exhaustive selection of the algorithm used to solve (1). We see that the analysis primal formulation proposes the best performance for each regularization parameter. We notice that the Condat (2013b) provides good performance too. Finally, the synthesis primal formulation along with the analysis dual formulation provides the slowest performance. Those results reinforces our choice to focus on the PGD of the analysis primal formulation.

# F    Proof for Section 2

## F.1    Convergence rate of PGD for the synthesis formulation (6)

*Proof.* The convergence rate of ISTA for the synthesis formulation reads

$$S(z^{(t)}) - S(z^*) \leq \frac{\widetilde{\rho}}{2t}\|z^{(0)} - z^*\|_2^2 \ . \tag{25}$$

(a) $\lambda = 0.1\lambda_{\max}$  (b) $\lambda = 0.8\lambda_{\max}$

Figure E.1: **Performance comparison** between the iterative solver for the synthesis and analysis formulation with the corresponding primal, dual or primal-dual re-parametrization. We notice that the primal analysis formulation provides the best performance in term of iterations. We also observe that the higher the regularization parameter, the faster the performance for each algorithm.

We use $S(z^{(t)}) = P(Lz^{(t)}) = P(u^{(t)})$ to get the correct left-hand side term. For the right hand side, we use $z^{(0)} = \widetilde{D}u^{(0)}$, and $z^* = \widetilde{D}u^*$, which gives $\|z^{(0)} - z^*\|_2^2 = \|\widetilde{D}(u^{(0)} - u^*)\|_2^2 \le 4\|u^{(0)} - u^*\|_2^2$. The last majoration comes from the fact that $\|\widetilde{D}\|_2^2 \le 4$, as shown per Lemma F.1. This yields

$$P(u^{(t)}) - P(u^*) \le \frac{2\widetilde{\rho}}{t}\|u^{(0)} - u^*\|_2^2 \ . \tag{26}$$

$\square$

## F.2   Computing the spectrum of $L$

**Lemma F.1.** *[Singular values of L] The singular values of $L \in \mathbb{R}^{k \times k}$ are given by*

$$\sigma_l = \frac{1}{2\cos(\frac{\pi l}{2k+1})}, \qquad \forall l \in \{1, \ldots, k\} \ .$$

*Thus,* $\|L\|_2 = \frac{2k+1}{\pi} + o(1)$.

*Proof.* As $L$ is invertible, so is $L^\top L$. To compute the singular values $\sigma_l$ of $L$, we will compute the eigenvalues $\mu_l$ of $(L^\top L)^{-1}$ and use the relation

$$\sigma_l = \frac{1}{\sqrt{\mu_l}} \tag{27}$$

With simple computations, we obtain

$$M_k = (L^\top L)^{-1} = L^{-1}(L^\top)^{-1} = \widetilde{D}\widetilde{D}^\top = \begin{bmatrix} 1 & -1 & 0 & \ldots & \\ -1 & 2 & -1 & 0 & \ldots \\ & \ddots & \ddots & \ddots & \\ & 0 & -1 & 2 & -1 \\ & & & -1 & 2 \end{bmatrix} \tag{28}$$

This matrix is tri-diagonal with a quasi-toepliz structure. Its characteristic polynomial $P_k(\mu)$ is given by:

$$P_k(\mu) = |\mu\,\mathrm{Id} - M_k| = \begin{vmatrix} \mu - 1 & 1 & 0 & \ldots & \\ 1 & \mu - 2 & 1 & 0 & \ldots \\ & \ddots & \ddots & \ddots & \\ 0 & & 1 & \mu - 2 & 1 \\ & 0 & & 1 & \mu - 2 \end{vmatrix} \tag{29}$$

$$= (\mu - 1)Q_{k-1}(\mu) - Q_{k-2}(\mu) \tag{30}$$

where (30) is obtained by developing the determinant relatively to the first line and $Q_k(\mu)$ is the characteristic polynomial of matrix $\widetilde{M_k}$ equal to $M_k$ except for the the top left coefficient which is replaced by 2

$$
\widetilde{M_k} = \begin{bmatrix} 2 & -1 & 0 & \dots & & \\ -1 & 2 & -1 & 0 & \dots & \\ & \ddots & \ddots & \ddots & & \\ & 0 & -1 & 2 & -1 \\ & & 0 & -1 & 2 \end{bmatrix} \tag{31}
$$

Using the same development as (30), one can show that $Q_k$ verifies the recurrence relationship

$$
Q_k(\mu) = (\mu - 2)Q_{k-1}(\mu) - Q_{k-2}(\mu); \qquad Q_1(\mu) = 2 - \mu, \quad Q_0(\mu) = 1 . \tag{32}
$$

Using this with (30) yields

$$
P_k(\mu) = Q_k(\mu) + Q_{k-1}(\mu) \tag{33}
$$

With the change of variable $\nu = \frac{\mu-2}{2}$ and denoting $U_k(\nu) = Q_k(2 + 2\nu)$, the recursion becomes

$$
U_k(\nu) = 2\nu U_{k-1}(\nu) - U_{k-2}(\nu); \qquad U_1(\nu) = 2\nu, \quad U_0(\mu) = 1 . \tag{34}
$$

This recursion defines the Chebyshev polynomials of the second kind (Chebyshev, 1853) which verifies the following relation

$$
\forall \theta \in [0, 2\pi], \quad U_k(\cos(\theta))\sin(\theta) = \sin((k+1)\theta) . \tag{35}
$$

Translating this relationship to $Q_k$ gives

$$
\forall \theta \in [0, 2\pi], \quad Q_k(2 + 2\cos(\theta))\sin(\theta) = \sin((k+1)\theta) . \tag{36}
$$

Using this in (33) shows that for $\theta \in [0, 2\pi[$ $P_k$ verify

$$
P_k(2 + 2\cos(\theta)\sin(\theta)) = \sin((k+1)\theta) + \sin(k\theta) . \tag{37}
$$

The equation

$$
\sin((k+1)\theta) + \sin(k\theta) = 0 , \tag{38}
$$

has $l$ solution in $[0, 2\pi[$ that are given by $\theta_l = \frac{2\pi l}{2k+1}$ for $l \in \{1, \dots n\}$. As for all $l$, $\sin(\theta_l) \neq 0$, the values $\mu_l = 2 + 2\cos(\theta_l) = 4\cos^2(\frac{\pi l}{2k+1})$ are the roots of $P_k$ and therefor the eigenvalues of $M_k$. Using (27) yields the expected value for $\sigma_l$.

The singular value of $L$ is thus obtain for $l = k$ and we get

$$
\|L\|_2 = \sigma_k = \frac{1}{2\cos(\frac{\pi k}{2k+1})} = \frac{1}{2\cos(\frac{\pi}{2}(1 - \frac{1}{2k+1}))} , \tag{39}
$$

$$
= \frac{1}{2\sin(\frac{\pi}{2}\frac{1}{2k+1})} = \frac{2k+1}{\pi} + o(1) . \tag{40}
$$

Where the last approximation comes from $\frac{1}{\sin(x)} = 1/x + o(1)$ when $x$ is close to 0. $\qquad\square$

### F.3 Proof for Proposition 2.1

**Proposition 2.1.** *[Lower bound for the ratio $\frac{\|AL\|_2^2}{\|A\|_2^2}$ expectation] Let $A$ be a random matrix in $\mathbb{R}^{m \times k}$ with i.i.d normally distributed entries. The expectation of $\|AL\|_2^2/\|A\|_2^2$ is asymptotically lower bounded when $k$ tends to $\infty$ by*

$$
\mathbb{E}\left[\frac{\|AL\|_2^2}{\|A\|_2^2}\right] \geq \frac{2k+1}{4\pi^2} + o(1)
$$

*Proof.* Finding the norm of $AL$ can be written as

$$
\|AL\|_2^2 = \max_{x \in \mathbb{R}^k} x L^\top A^\top A L x; \quad s.t. \ \|x\|_2 = 1 \tag{41}
$$

From Lemma F.1, we can write $L = W^\top \Sigma V$ with $V$, $W$ two unitary matrices and $\Sigma$ a diagonal matrix with $\Sigma_{l,l} = \sigma_l$ for all $l \in \{1, .., k\}$.

First, we consider the case where $A^\top A$ is a rank one matrix with $A^\top A = \|A\|_2^2 u_1 u_1^\top$, with vector $u_1$ uniformly sampled from the $\ell_2$-ball and fixed $\|A\|_2$. In this case, as $W$ is unitary, $w_1 = W u_1$ is also a vector uniformly sampled from the sphere. Also as $V$ is unitary, it is possible to re-parametrize (41) by $y = Vx$ such that

$$\max_{y \in \mathbb{R}^k} \|A\|_2^2 y^\top \Sigma w_1 w_1^\top \Sigma y; \quad s.t. \ \|y\|_2 = 1 \tag{42}$$

This problem can be maximized by taking $y = \frac{\Sigma u_1}{\|\Sigma u_1\|_2}$, which gives

$$\|AL\|_2^2 = \|A\|_2^2 \|\Sigma w_1\|_2^2 \tag{43}$$

Then, we compute the expectation of $\|\Sigma w_1\|_2^2$ with respect with $w_1$, a random vector sampled in the $\ell_2$ unit ball,

$$\mathbb{E}_{w_1}[\|\Sigma w_1\|_2^2] = \sum_{l=1}^{k} \sigma_l^2 \mathbb{E}[u_{1,i}^2] = \sum_{l=1}^{k} \frac{1}{4\cos^2 \frac{\pi l}{2k+1}} \frac{1}{k} = \frac{1}{2\pi} \sum_{l=1}^{k} \frac{\pi}{2k} \frac{1}{\cos^2 \frac{\pi l}{2k+1}} \ . \tag{44}$$

Here, we made use of the fact that for a random vector $u_1$ on the sphere in dimension $k$, $\mathbb{E}[u_{1,i}] = \frac{1}{k}$ In the last part of the equation, we recognize a Riemann sum for the interval $[0, \frac{\pi}{2}[$. However, $x \mapsto \frac{1}{\cos^2(x)}$ is not integrable on this interval. As the function is positive and monotone, we can still use the integral to highlight the asymptotic behavior of the series. For $k$ large enough, we consider the integral

$$\int_0^{\frac{\pi}{2} - \frac{\pi}{2k+1}} \frac{1}{\cos^2(x)} dx = \left[\frac{sin(x)}{\cos(x)}\right]_0^{\frac{\pi}{2} - \frac{\pi}{2k+1}} = \frac{\cos \frac{\pi}{2k+1}}{\sin \frac{\pi}{2k+1}} = \frac{2k+1}{\pi} + o(1) \tag{45}$$

Thus, for $k$ large enough, we obtain

$$\mathbb{E}_{w_1}\left[\|\Sigma w_1\|_2^2\right] = \frac{1}{2\pi}\left(\frac{2k+1}{\pi} + o(1)\right) \tag{46}$$

Thus, we get

$$\mathbb{E}\left[\frac{\|AL\|_2^2}{\|A\|_2^2}\right] = \left(\frac{k + \frac{1}{2}}{\pi^2} + o(1)\right) \tag{47}$$

This concludes the case where $A^\top A$ is of rank-1 with uniformly distributed eigenvector.

In the case where $A^\top A$ is larger rank, it is lower bounded by $\|A\|_2^2 u_1 u_1^\top$ where $u_1$ is its eigenvector associated to its largest eigenvalue, since it is *psd*. Since $A^\top A$ is a Whishart matrix, its eigenvectors are uniformly distributed on the sphere (Silverstein, 1989). We can thus use the same lower bound as previously for the whole matrix.

$\square$

# G   Proof for Section 3

## G.1   Proof for Proposition 3.1

**Proposition 3.1.** *[Weak Jacobian of prox-TV] Let $x \in \mathbb{R}^k$ and $u = prox_{\mu\|\cdot\|_{TV}}(x)$, and denote by $\mathcal{S}$ the support of $z = \widetilde{D}u$. Then, the weak Jacobian $J_x$ and $J_\mu$ of the prox-TV relative to $x$ and $\mu$ can be computed as*

$$J_x(x, \mu) = L_{:,\mathcal{S}}(L_{:,\mathcal{S}}^\top L_{:,\mathcal{S}})^{-1} L_{:,\mathcal{S}}^\top \quad and \quad J_\mu(x, \mu) = -L_{:,\mathcal{S}}(L_{:,\mathcal{S}}^\top L_{:,\mathcal{S}})^{-1} \text{sign}(Du)_{\mathcal{S}}$$

First, we recall Lemma G.1 to weakly derive the soft-thresholding.

**Lemma G.1** (Weak derivative of the soft-thresholding; Deledalle et al. 2014)**.** *The soft-thresholding operator $ST : \mathbb{R} \times \mathbb{R}_+ \mapsto \mathbb{R}$ defined by $ST(t, \tau) = \text{sign}(t)(|t| - \tau)_+$ is weakly differentiable with weak derivatives*

$$\frac{\partial ST}{\partial t}(t, \tau) = \mathbb{1}_{\{|t|>\tau\}} \ , \quad and \quad \frac{\partial ST}{\partial \tau}(t, \tau) = -\text{sign}(t) \cdot \mathbb{1}_{\{|t|>\tau\}} \ ,$$

*where*

$$\mathbb{1}_{\{|t|>\tau\}} = \begin{cases} 1, & if \ |t| > \tau, \\ 0, & otherwise. \end{cases}$$

A very important remark here is to notice that if one denote $z = \mathrm{ST}(t, \tau)$, one can rewrite these weak derivatives as

$$\frac{\partial\,\mathrm{ST}}{\partial t}(t, \tau) = \mathbb{1}_{\{|z|>0\}} \;, \qquad \text{and} \qquad \frac{\partial\,\mathrm{ST}}{\partial \tau}(t, \tau) = -\operatorname{sign}(z) \cdot \mathbb{1}_{\{|z|>0\}} \;. \tag{48}$$

Indeed, when $|t| > \tau$, $|z| = |t| - \tau > 0$ and conversely, $|z| = 0$ when $|t| < \tau$. Moreover, when $|t| > \tau$, we have $\operatorname{sign}(t) = \operatorname{sign}(z)$ and thus the two expressions for $\frac{\partial\,\mathrm{ST}}{\partial \tau}$ match.

Using this Lemma G.1, we now give the proof of Proposition 3.1.

*Proof.* The proof is inspired from the proof from Bertrand et al. (2020, Proposition 1). We denote $u(x, \mu) = \operatorname{prox}_{\mu\|\cdot\|_{TV}}(x)$, hence $u(x, \mu)$ is defined by

$$u(x, \mu) = \arg\min_{\hat{u}} \frac{1}{2}\|x - \hat{u}\|_2^2 + \mu\|\hat{u}\|_{TV} \tag{49}$$

Equivalently, as we have seen previously in (5), using the change of variable $\hat{u} = L\hat{z}$ and minimizing over $\hat{z}$ gives

$$\min_{\hat{z}} \frac{1}{2}\|x - L\hat{z}\|_2 + \mu\|R\hat{z}\|_1 \;. \tag{50}$$

We denote by $z(x, \mu)$ the minimizer of the previous equation. Thus, the solution $u(x, \mu)$ of the original problem (49) can be recovered using $u(, \mu) = Lz(x, \mu)$. Iterative PGD can be used to solve (50) and $z(x, mu)$ is a fixed point of the iterative procedure. That is to say the solution $z$ verifies

$$\begin{cases} z_1(x, \mu) = z_1(x, \mu) - \frac{1}{\rho}(L^\top(Lz(x, \mu) - x))_1 \;, \\ z_i(x, \mu) = \mathrm{ST}\left(z_i(x, \mu) - \frac{1}{\rho}(L^\top(Lz(x, \mu) - x))_i, \frac{\mu}{\rho}\right) \quad \text{for } i = 2\ldots k \;. \end{cases} \tag{51}$$

Using the result from Lemma G.1, we can differentiate (51) and obtain the following equation for the weak Jacobian $\widehat{J}_x(x, \mu) = \frac{\partial z}{\partial x}(x, \mu)$ of $z(x, \mu)$ relative to $x$

$$\widehat{J}_x(x, \mu) = \begin{pmatrix} 1 \\ \mathbb{1}_{\{|z_2(x,\mu)|>0\}} \\ \vdots \\ \mathbb{1}_{\{|z_k(x,\mu)|>0\}} \end{pmatrix} \odot \left[ (\mathrm{Id} - \frac{1}{\rho}L^\top L)\widehat{J}_x(x, \mu) + \frac{1}{\rho}L^\top \,\mathrm{Id} \right] \;. \tag{52}$$

Identifying the non-zero coefficient in the indicator vectors yields

$$\begin{cases} \widehat{J}_{x,\mathcal{S}^c}(x, \mu) = 0 \\ \widehat{J}_{x,\mathcal{S}}(x, \mu) = (\mathrm{Id} - \frac{1}{\rho}L_{:,\mathcal{S}}^\top L_{:,\mathcal{S}})\widehat{J}_{x,\mathcal{S}}(x, \mu) + \frac{1}{\rho}L_{:,\mathcal{S}}^\top \;. \end{cases} \tag{53}$$

As, $L$ is invertible, so is $L_{:,\mathcal{S}}^\top L_{:,\mathcal{S}}$ for any support $\mathcal{S}$ and solving the second equations yields the following

$$\widehat{J}_{x,\mathcal{S}} = (L_{:,\mathcal{S}}^\top L_{:,\mathcal{S}})^{-1}L_{:,\mathcal{S}}^\top \tag{54}$$

Using $u = Lz$ and the chain rules yields the expecting result for the weak Jacobian relative to $x$, noticing that as $\hat{J}_{x,\mathcal{S}^c} = 0$, $L\hat{J}_x = L_{:,\mathcal{S}}\hat{J}_{x,\mathcal{S}}$.

Similarly, concerning, $\widehat{J}_\mu(x, \mu)$, we use the result from Lemma G.1 an differentiale (51) and obtain $\widehat{J}_\mu(x, \mu) = \frac{\partial z}{\partial \mu}(x, \mu)$ of $z(x, \mu)$ relative to $\mu$

$$\widehat{J}_\mu(x, \mu) = \begin{pmatrix} 1 \\ \mathbb{1}_{\{|z_2(x,\mu)|>0\}} \\ \vdots \\ \mathbb{1}_{\{|z_k(x,\mu)|>0\}} \end{pmatrix} \odot \left[ (\mathrm{Id} - \frac{1}{\rho}L^\top L)\widehat{J}_\mu(x, \mu) \right] + \frac{1}{\rho}\begin{pmatrix} 1 \\ -\operatorname{sign}(z_2(x,\mu))\mathbb{1}_{\{|z_2(x,\mu)|>0\}} \\ \vdots \\ -\operatorname{sign}(z_k(x,\mu))\mathbb{1}_{\{|z_k(x,\mu)|>0\}} \end{pmatrix} \;. \tag{55}$$

Identifying the non-zero coefficient in the indicator vectors yields

$$\begin{cases} \widehat{J}_{\mu,\mathcal{S}^c}(x,\mu) &= 0 \\ \widehat{J}_{\mu,\mathcal{S}}(x,\mu) &= \widehat{J}_{\mu,\mathcal{S}^c}(x,\mu) - \frac{1}{\rho}L_{:,\mathcal{S}}^\top L_{:,\mathcal{S}}\widehat{J}_{\mu,\mathcal{S}^c}(x,\mu) - \frac{1}{\rho}\operatorname{sign}(z_S(x,\mu)) \ . \end{cases} \tag{56}$$

As previous, solving the second equation yields the following

$$\widehat{J}_{\mu,\mathcal{S}} = -(L_{:,\mathcal{S}}^\top L_{:,\mathcal{S}})^{-1}\operatorname{sign}(z_S(x,\mu)) \tag{57}$$

Using $u = Lz$ and the chain rules yields the expecting result for the weak Jacobian relative to $\mu$, noticing that as $\hat{J}_{\mu,\mathcal{S}^c} = 0$.

$\square$

## G.2 Convergence of the weak Jacobian

**Proposition G.2.** *Linear convergence of the weak Jacobian We consider the mapping* $z^{(T_i n)}$ : *,$\mu\mathbb{R}^k \times \mathbb{R}_+ \mapsto \mathbb{R}^k$ defined where $z^{(T_i n)}(x)$ is defined by recursion*

$$z^{(t)}(x,\mu) = ST(z^{(t-1)}(x,\mu) - \frac{1}{\|L\|_2^2}L^\top(Lz^{(t-1)}(x,\mu) - x), \frac{\mu}{\|L\|_2^2} \ . \tag{58}$$

*Then the weak $\mathcal{J}_x = L\frac{\partial z^{(T_i n)}}{\partial x}$ and $\mathcal{J}_\mu = L\frac{\partial z^{(T_i n)}}{\partial \mu}$ of this mapping relative to the inputs $x$ and $\mu$ converges linearly toward the weak Jacobian $J_x$ and $J_\mu$ of $prox_{\mu\|\cdot\|_{TV}}(x)$ defined in* [Proposition 3.1](#).

This mapping defined in [(58)](#) corresponds to the inner network in LPGD-LISTA when the weights of the network have not been learned.

*Proof.* As $L$ is invertible, problem [(50)](#) is strongly convex and have a unique solution. We can thus apply the result from [Bertrand et al. (2020)](#), Proposition 2) which shows the linear convergence of the weak Jacobian $\hat{\mathcal{J}}_x = \frac{\partial z^{(T_i n)}}{\partial x}$ and $\hat{\mathcal{J}}_\mu = \frac{\partial z^{(T_i n)}}{\partial \mu}$ for ISTA toward $\hat{J}_x$ and $\hat{J}_\mu$ of the synthesis formulation of the prox. Using the linear relationship between the analysis and the synthesis formulations yields the expected result. $\square$

## G.3 Estimating $T_{in}$ and $T$ to achieve $\delta$ error

Using inexact proximal gradient descent results from [Schmidt et al. (2011)](#) and [Machart et al. (2012)](#), we compute the scaling of $T_{in}$ and $T$ to achieve a given error level $\delta > 0$.

**Proposition G.3.** *[Scaling of $T$ and $T_{in}$ w.r.t the error level $\delta$] Let $\delta$ the error defined such as $P_x(u^{(T)}) - P_x(u^*) \leq \delta$.*
*We suppose there exists some constants $C_0 \geq \|u^{(0)} - u^*\|_2$ and $C_1 \geq \max_\ell \|u^{(\ell)} - prox_{\frac{\mu}{\rho}}(u^{(\ell)})\|_2$*
*Then, $T$ the number of layers for the global network and $T_{in}$ the inner number of layers for the prox-TV scale are given by*

$$T_{in} = \frac{\ln \frac{1}{\delta} + \ln 6\sqrt{2\rho}C_1}{\ln \frac{1}{1-\gamma}} \quad and \quad T = \frac{2\rho C_0^2}{\delta}$$

*with $\rho$ defined as in* [(2)](#)

*Proof.* As discussed by [Machart et al. (2012)](#), the global convergence rate of inexact PGD with $T_{in}$ inner iteration is given by

$$P_x(u^{(T)}) - P_x(u^*) \leq$$

$$\frac{\rho}{2T}\left(\|u^{(0)} - u^*\|_2 + 3\sum_{\ell=1}^T \sqrt{\frac{2(1-\gamma)^{T_{in}}\|u^{(\ell-1)} - prox_{\frac{\mu}{\rho}}(u^{(\ell-1)})\|_2^2}{\rho}}\right)^2 , \tag{59}$$

where $\gamma$ is the condition number for $L$ *i.e.* $\frac{\cos(\frac{\pi}{2k+1})}{\sin(\frac{\pi}{2k+1})}$.

We are looking for minimal parameters $T$ and $T_{in}$ such that the error bound in (59) is bellow a certain error level $\delta$.

We consider the case where there exists some constants $C_0 \geq \|u^{(0)} - u^*\|_2$ and $C_1 \geq \max_\ell \|u^{(\ell)} - \text{prox}_{\frac{\mu}{\rho}}(u^{(\ell)})\|_2$ upper bounding how far the initialization can be compared to the result of the global problem and the sub-problems respectively.

We denote $\alpha_1 = 3\sqrt{\frac{2}{\rho}}C_1$. The right hand side of (59) can be upper bounded by as

$$\frac{\rho}{2T}\left(\|u^{(0)} - u^*\|_2 + 3\sum_{\ell=1}^{T}\sqrt{\frac{2(1-\gamma)^{T_{in}}\|u^{(\ell-1)} - \text{prox}_{\frac{\mu}{\rho}}(u^{(\ell-1)})\|_2^2}{\rho}}\right)^2 \tag{60}$$

$$\leq \frac{\rho}{2T}\left(C_0 + \alpha_1 T(1-\gamma)^{T_{in}/2}\right)^2$$

Then, we are looking for $T, T_{in}$ such that this upper bound is lower than $\delta$, *i.e.*

$$\frac{\rho}{2T}\left(C_0 + \alpha_1 T(1-\gamma)^{T_{in}/2}\right)^2 \leq \delta \tag{61}$$

$$\Leftrightarrow \left(C_0 + \alpha_1 T(1-\gamma)^{T_{in}/2}\right)^2 - \frac{2\delta}{\rho}T \leq 0 \tag{62}$$

$$\Leftrightarrow \left(C_0 + \alpha_1 T(1-\gamma)^{T_{in}/2} - \sqrt{\frac{2\delta}{\rho}}\sqrt{T}\right)\underbrace{\left(B + \alpha_1 T(1-\gamma)^{T_{in}/2} + \sqrt{\frac{2\delta}{\rho}}\sqrt{T}\right)}_{\geq 0} \leq 0 \tag{63}$$

$$\Leftrightarrow C_0 + \alpha_1 T(1-\gamma)^{T_{in}/2} - \sqrt{\frac{2\delta}{\rho}}\sqrt{T} \leq 0 \tag{64}$$

$$\tag{65}$$

Denoting $\alpha_2 = \sqrt{\frac{2\delta}{\rho}}$ and $X = \sqrt{T}$, we get the following function of $X$ and $T_{in}$

$$f(X, T_{in}) = \alpha_1(1-\gamma)^{T_{in}/2}X^2 - \alpha_2 X + C_0 \tag{66}$$

The inequality $f(X, T_{in}) \leq 0$ has a solution if and only if $\alpha_2^2 - 4C_0\alpha_1(1-\gamma)^{T_{in}/2} \geq 0$ *i.e.*

$$T_{in} \geq 2\frac{\ln\frac{\alpha_2^2}{4\alpha_1 C_0}}{\ln 1 - \gamma}$$

Taking the minimal value for $T_{in}$ *i.e.* $T_{in} = 2\frac{\ln\frac{\alpha_2^2}{4\alpha_1 C_0}}{\ln 1-\gamma} = \frac{\ln\frac{1}{\delta} + \ln 6\sqrt{2\rho}C1}{\ln\frac{1}{1-\gamma}}$ yields

$$f(X, T_{in}) = \frac{\alpha_2^2}{4C_0}X^2 - \alpha_2 X + C_0 = \frac{\alpha_2^2}{4C_0}(X - \frac{2C_0}{\alpha_2})^2$$

for $X = \frac{2C_0}{\alpha_2} = \frac{\sqrt{2\rho}C_0}{\sqrt{\delta}}$ *i.e.* $T = \frac{2\rho C_0^2}{\delta}$. $\qquad\square$