[Reviews · NeurIPS 2020]

Review 1

Summary and Contributions: The paper considers the problem of accelerating TV-regularized problems. The paper first shows that assuming random Gaussian design, the analysis formulation might converge faster than the synthesis formulation based on the usual convergence rate for PGD. The paper then proposes two methods to accelerate the analysis formulation, using unrolling as done in LISTA. With regular lasso, the proximal operator is soft thresholding and back propagation is easy. With TV, backpropagation is a bit more difficult, and the paper proposes two alternatives. Numerical experiments are performed both on synthetic and real fMRI data, where the methods are compared.

Strengths: TV regularization is a problem of interest to the community, and the proposed methods are interesting. In particular, using LISTA internally seems to lead to much better performance in low sparsity regimes. Code is available as part of a general toolbox for TV-regularized problems, which makes it easy to use and disseminate.

Weaknesses: The contribution is mostly a performance improvement in terms of number of iterations/run time to solve the TV-regularized inverse problems. I am not familiar with applications enough to assess whether this would have a significant impact.

Correctness: The paper appears correct overall.

Clarity: The paper is clear and well written. My only (minor) concern is at L80: "In the 1D case" seems to imply that the paper considers general differential operators, but L67 introduces `D` as the 1D finite differences operator. This is mentioned again L166. Proposition 3.1 while Section 3.2 seems to be able to support more than, but the presentation does not make that very clear in my opinion. Minor typos: - L221: "the between the" - L274: "as mentionED"?

Relation to Prior Work: The paper cites prior work on accelerating TV-regularized problems as well as some applications.

Reproducibility: Yes

Additional Feedback:


Review 2

Summary and Contributions: The paper proposes two unrolled algorithms based on proximal gradient descent and LISTA network for solving the total variation regularized signal recovery problem. Numerical experiments on 1D synthetic and real data have justified the proposed effectiveness in terms of convergence speed.

Strengths: Combination of neural networks and the classical proximal gradient descent algorithms is the major strength of the work, which sounds interesting and important to other related works.

Weaknesses: Theoretical discussion of the proposed algorithms seems to be still at an early development stage without any convergence or stability guarantee. Section 2.1 seems to review the synthesis formulation with the proposed proposition and conjecture, which however do not shed lights on the later empirical evaluations. Thus the theoretical contributions of this work are not clear. Furthermore, similarity of the training data and the test data plays a crucial role in neural network type of deep learning algorithms while there is no such a restriction in the traditional gradient descent and its iterative variants, which may limit the performance of the proposed algorithms in some circumstances.

Correctness: 1. The claims and method are mostly correct but not always clear enough. In Proposition 3.1, a unique form for the Jacobian $J_\mu$ seems to disagree with the fact that the prox-TV involves the non-smooth TV term. Is subgradient considered here? 2. Numerical experiments on classical image recovery applications, e.g., image denoising and deconvolution, could be conducted, which would bring more straightforward performance comparisons.

Clarity: Mostly but not always. But there are some minor language issues, e.g., "close-form"->"closed-form" and "one need"->"one needs" in line 79. Notation is not always consistent as well, e.g., (4) and (6) use P while (1) uses P_x which should be either defined separately or corrected to be consistent. In proposition 2.1, does "normal entries" mean by "normally distributed entries"?

Relation to Prior Work: Yes.

Reproducibility: No

Additional Feedback: In the abstract, the motivation and contributions of the proposed works should be briefly introduced and highlighted. In lines 80-83, alternating direction method of multipliers (ADMM) and its variants including Nesterov accelerated versions are not reviewed.


Review 3

Summary and Contributions: In recent years, there has been extensive research into unrolling optimization algorithms (developed for imaging or inverse problems) as neural networks and learning the parameters. However, much of the existing works are in the synthesis regime (where they unroll for example PGD), the contributions of this work is to show how to unroll in the analysis regime, and in particular, with a TV regulariser. The contributions is essentially applying LISTA for TV regularized problems, with an approximate prox. The results are largely empirical, and the proposed schemes apply only to the 1D setting. The authors propose to compute the TV-prox using either the Taut string scheme of Condat, or by reformulating the prox problem as an lasso problem and applying LISTA. After reading the author response, I am inclined to stay with my initial assessment that this is 'marginally above the acceptance threshold'. This paper's novelty of dealing with the prox is somewhat limited because this work restricts to the 1D TV case, where it is well known (due to the work of Condat) that one can compute the prox in a fast and non-iterative manner.

Strengths: This is mainly an empirical study, they argue that the approach of reformulating an analysis problem as a synthesis problem (even if this can be done) will lead to suboptimal performance. It is therefore necessary to approximate prox-TV directly, and the authors demonstrate some computation gain in doing so.

Weaknesses: - their results only apply to 1D TV. Of course, the most natural one when working beyond 1D is to compute the prox by solving the dual problem and this can be done using PGD, however, this is not investigated, but mentioned briefly in the conclusion. - They show and comment in the numerics that there is little gain between simply applying PGD to solve this lasso problem vs LISTA. Hence, they replace the prox term of LISTA with very standard approximate prox techniques. - Experimental setup and precise methodology is unclear: There are no details on the training procedure. Do you train the weights for LISTA approximation of the prox term separately? - If one wants an accurate solution, then it seems from fig. 3 that accelerated PGD performs just as well as LPGD-Taut in left plot and better than both learned methods in the right plot. So, despite the high computational effort in training a network in the first place, it seems that the computational gains are actually quite modest? You show error per layer, am I right in assuming that the computational cost for each layer is the same across the different methods? - They show and comment in the numerics that there is little gain between simply applying PGD to compute prox_TV vs LISTA (Figure 4). So, it seems likely that the runtime computation time may be dominance by the prox calculation and there will be limited computational gain overall,

Correctness: The claims are appropriate.

Clarity: Yes.

Relation to Prior Work: Contributions and previous works are clearly described.

Reproducibility: Yes

Additional Feedback:


Review 4

Summary and Contributions: This paper proposes two approaches to efficiently compute the derivatives for TV-regularised problems via learned proximal gradient descent using deep neural networks to emulate the algorithm's iterations. It analyzes their benefits and limitations and discusses the regime in which the proposed approaches can improve over the iterative analogues.

Strengths: * The paper is well structured. There is a clear motivation given, well documented related work and experiments demonstrating the claims. The proposed approaches are clear in comparison and context of the related work. * The theoretical claims are well justified, with proofs provided in the Supplementary Material. Code implementing the described approaches is also provided. One can reproduce the shown results. * Experiments demonstrate that the learned networks for prox-TV provide a significant advantage in convergence speed.

Weaknesses: * The novelty of the work is not that significant. Using deep nets to model unrolled algorithms has been proposed already in the methods by Gregor and LeCun (2010), which are described in the related works. The novelty lies in formulating the algorithms in their "analysis" instead of their "synthesis" version. Further, this paper only applies to 1D TV problems. * Many of the equations in the paper mix / equate the minimizer with the objective or the minimum with the objective. It is possible that the notation is such in order to save space, but it is nevertheless incorrect and might confuse the readers. * A few grammatical errors here and there, mostly using singular instead of plural.

Correctness: Yes.

Clarity: Yes.

Relation to Prior Work: Yes.

Reproducibility: Yes

Additional Feedback:

[Author Response · NeurIPS 2020]

We would like to thank the reviewers for their encouraging and instructive comments. We also thanks the reviewers for pointing out typos and inconsistency in the notations that will be corrected in the paper. We first address generals concerns shared by the reviewers and then give detailed answers to other points raised by each reviewer.

**The work is limited to the 1D case.** We restricted our attention to the 1D case for two reasons: 1) It is not trivial to propose an analysis formulation for a network even in the 1D case as this requires deriving through the prox - which represents one of our main contribution - and we showed that these development are necessary due to the inefficiency of synthesis based approaches. This is a significant necessary first step to problems in 2D/3D. 2) The extension to larger dimensional signals requires further development that would complexify the paper and make the comparison less clear. For the 2D case, we believe we can use similar tools using equivalence with LASSO using synthesis formulation. However, this would require a projection on the tangent space of the integrable discrete signals, which we feel is out of scope for this paper. The path mentioned in the conclusion and by R#4 of using the dual might also be an alternative but shifting from the dual space to the primal without losing the first order information is again non-trivial. Both these extensions share the same basics as our proposed method but require more tools (duality or manifold analysis) so we consider it is out of scope for this paper.

**Reproducibility:** We thanks reviewers R#3 & R#4 for highlighting that some details of our experimental setup are unclear in the text. To answer R#4, the weights of the LISTA approximation of the prox are trained jointly. We will clarify this in the paper and also add a section in appendix to further detail the training procedure used.

**R#2:** **The contribution is mostly a performance improvement.** We would like to point to the reviewer that our contribution is centered on proposing a learning based acceleration for TV-regularized problems by relying on an analysis formulation. To the best of our knowledge, this has not been explored before, and it is not trivial as it requires the derivation through a prox. As a consequence, one can obtain performance improvements, though we envision our contribution impacting several other problems where analysis-based regularization techniques are important. Some of these include applications to functional MRI as mentioned in the paper but also pre-processing of physiological signals for health care and analysis of sensor data in industrial context. Moreover, our work also gives a way to compute the weak derivative of TV-regularized problems, which could be used for hyper-parameter $\lambda$ tuning or for dictionary learning methods.

**R#3:** **Theoretical contributions of this work are not clear.** While it is true that many questions of convergence remain open, our theoretical results are important because they depict the central role of optimizing the original analysis formulation. One might think that we can deploy the original synthesis formulation by directly applying LISTA to learn a network. However this is found to be slow (see Fig.3). Our theoretical analysis explains *why* this is the case, and as a result why it is important to develop such network architecture for the analysis formulation instead.

**Generalizability of the learned network.** This is true for any approach that relies on learned algorithms, and indeed most statistical learning methods rely on training samples to be sampled from the same distribution as those at test time. In our context, the key observation is that most of the time, one does not care about solving this problem for any signal in the entire space but rather only those in a subset of such space. Additionally, using learned algorithms can accelerate the optimization of the loss for a fixed number of iterations. The whole point in learning such a network is that one can provide faster approximate convergence for signals coming from the *same distribution* as your training set with a *fixed number of iterations/layers*.

**Uniqueness of the Jacobian** This is indeed a weak Jacobian and not the Jacobian. We will clarify accordingly.

**Abstract and related work** We will improve the abstract to better mention our contributions and also mention ADMM in methods to solve TV regularized problems.

**R#4:** **Comparison with accelerated PGD.** The computational gain is significant for solutions with some approximation error, i.e. where few layers or iterations are used. If one is after exact solutions, then non-learned solvers are to be employed. However, learned approaches can be very useful in practice as one is often interested in approximated solutions with controlled tolerances. Also, while it is true that accelerated PGD is faster at the end, learned approaches are faster non accelerated PGD. Note that the same acceleration mechanisms could also be used in the network, similarly to LFISTA developed in Moreau & Bruna (2017). However, we chose not to include this as it increases the complexity of the architecture and makes it less clear for comparison purposes.

**Computational cost for each layer is the same** This is correct. In particular, the iterative version of an algorithm has the exact same computational cost as the one of a learned network based on this method at test time.

**Limited computational gain overall.** Figure 4 depicts the value of the proximal loss, F(z), in Eq. (3). This shows that our learned approach achieves a slightly lower error on the prox than directly applying ISTA. When combined in the overall algorithm, while the quality of the prox does not change a lot, this leads to a large computational gain, as illustrated in Fig.3.

**R#5:** **The novelty of the work is not that significant.** The formulation of a learnable network in the analysis instead of the synthesis (LISTA) is not trivial as it requires to differentiate the prox operator, and this has prevented the application of these ideas to this kind of problems. To the best of our knowledge, we are the first to apply deep nets to accelerate methods that contain a non separable prox which are present in many applications.

[Meta-Review · NeurIPS 2020]

This paper proposes algorithms for solving the total variation regularized signal recovery problem. The proposed approach is interesting with theoretical justifications. The reviewers provide many constructive comments that should be addressed in the final version of the paper.